# Combining Ensembles and Data Augmentation can Harm your Calibration

**Yeming Wen**[*1], **Ghassen Jerfel**[*2], **Rafael Muller**[2], **Michael W. Dusenberry**[2],
**Jasper Snoek**[2], **Balaji Lakshminarayanan**[2] & **Dustin Tran**[2]
* Equal contribution, [1]University of Texas, Austin, [2]Google Brain

## Abstract

Ensemble methods which average over multiple neural network predictions are a simple approach to improve a model's calibration and robustness. Similarly, data augmentation techniques, which encode prior information in the form of invariant feature transformations, are effective for improving calibration and robustness. In this paper, we show a surprising pathology: combining ensembles and data augmentation can harm model calibration. This leads to a trade-off in practice, whereby improved accuracy by combining the two techniques comes at the expense of calibration. On the other hand, selecting only one of the techniques ensures good uncertainty estimates at the expense of accuracy. We investigate this pathology and identify a compounding under-confidence among methods which marginalize over sets of weights and data augmentation techniques which soften labels. Finally, we propose a simple correction, achieving the best of both worlds with significant accuracy and calibration gains over using only ensembles or data augmentation individually. Applying the correction produces new state-of-the art in uncertainty calibration across CIFAR-10, CIFAR-100, and ImageNet.[1]

## 1 Introduction

Many success stories in deep learning (Krizhevsky et al., 2012; Sutskever et al., 2014) are in restricted settings where predictions are only made for inputs similar to the training distribution. In real-world scenarios, neural networks can face truly novel data points during inference, and in these settings it can be valuable to have good estimates of the model's uncertainty. For example, in healthcare, reliable uncertainty estimates can prevent over-confident decisions for rare or novel patient conditions (Dusenberry et al., 2019). We highlight two recent trends obtaining state-of-the-art in uncertainty and robustness benchmarks.

Ensemble methods are a simple approach to improve a model's calibration and robustness (Lakshminarayanan et al., 2017). The same network architecture but optimized with different initializations can converge to different functional solutions, leading to decorrelated prediction errors. By averaging predictions, ensembles can rule out individual mistakes (Lakshminarayanan et al., 2017; Ovadia et al., 2019). Additional work has gone into efficient ensembles such as MC-dropout (Gal and Ghahramani, 2016), BatchEnsemble, and its variants (Wen et al., 2020; Dusenberry et al., 2020; Wenzel et al., 2020). These methods significantly improve calibration and robustness while adding few parameters to the original model.

Data augmentation is an approach which is orthogonal to ensembles in principle, encoding additional priors in the form of invariant feature transformations. Intuitively, data augmentation enables the model to train on more data, encouraging the model to capture certain invariances with respect to its inputs and outputs; data augmentation may also produce data that may be closer to an out-of-distribution target task. It has been a key factor driving state-of-the-art: for example, Mixup (Zhang et al., 2018; Thulasidasan et al., 2019a), AugMix (Hendrycks et al., 2020), and test-time data augmentation (Ashukha et al., 2020).

A common wisdom in the community suggests that ensembles and data augmentation should naturally combine. For example, the majority of uncertainty models in vision with strong performance are

---

[1]Contact: ywen@utexas.edu. Code: `https://github.com/google/edward2/tree/master/experimental/marginalization_mixup`.

built upon baselines leveraging standard data augmentation (He et al., 2016; Hendrycks et al., 2020) (e.g., random flips, cropping); Hafner et al. (2018) cast data augmentation as an explicit prior for Bayesian neural networks, treating it as beneficial when ensembling; and Hendrycks et al. (2020) highlights further improved results in AugMix when combined with Deep Ensembles (Hansen and Salamon, 1990; Krogh and Vedelsby, 1995). However, we find the complementary benefits between data augmentations and ensembels are not universally true. Section 3.1 illustrates the poor calibration of combining ensembles (MC-dropout, BatchEnsemble and Deep Ensembles) and Mixup on CIFAR: the model outputs excessive low confidence. Motivated by this pathology, in this paper, we investigate in more detail why this happens and propose a method to resolve it.

**Contributions.** In contrast to prior work, which finds individually that ensembles and Mixup improve calibration, we find that combining ensembles and Mixup consistently degrades calibration performance across three ensembling techniques. From a detailed analysis, we identify a compounding under-confidence, where the soft labels in Mixup introduce a negative confidence bias that hinders its combination with ensembles. We further find this to be true for other label-based strategies such as label smoothing. Finally, we propose CAMixup to correct this bias, pairing well with ensembles. CAMixup produces new *state-of-the-art calibration* on both CIFAR-10/100 (e.g., $0.4\%$ and $2.3\%$ on CIFAR-10 and CIFAR-10C), building on Wide ResNet 28-10 for competitive accuracy (e.g., $97.5\%$ and $89.8\%$) and on ImageNet ($1.5\%$), building on ResNet-50 for competitive accuracy ($77.4\%$).

## 2 BACKGROUND ON CALIBRATION, ENSEMBLES AND DATA AUGMENTATION

### 2.1 CALIBRATION

Uncertainty estimation is critical but ground truth is difficult to obtain for measuring performance. Fortunately, calibration error, which assesses how well a model reliably forecasts its predictions over a population, helps address this. Let $(\hat{Y}, \hat{P})$ denote the class prediction and associated confidence (predicted probability) of a classifier.

**Expected Calibration Error(ECE)**: One notion of miscalibration is the expected difference between confidence and accuracy (Naeini et al., 2015): $E_{\hat{P}}[|\mathbb{P}(\hat{Y} = Y|\hat{P} = p) - p|]$. ECE approximates this by binning the predictions in $[0, 1]$ under $M$ equally-spaced intervals, and then taking a weighted average of each bins' accuracy/confidence difference. Let $B_m$ be the set of examples in the $m^{th}$ bin whose predicted confidence falls into interval $(\frac{m-1}{M}, \frac{m}{M}]$. The bin $B_m$'s accuracy and confidence are:

$$\text{Acc}(B_m) = \frac{1}{|B_m|} \sum_{x_i \in B_m} \mathbb{1}(\hat{y}_i = y_i), \quad \text{Conf}(B_m) = \frac{1}{|B_m|} \sum_{x_i \in B_m} \hat{p}_i, \qquad (1)$$

where $\hat{y}_i$ and $y_i$ are the predicted and true labels and $\hat{p}_i$ is the confidence for example $x_i$. Given $n$ examples, ECE is $\sum_{m=1}^{M} \frac{|B_m|}{n} \left| \text{Acc}(B_m) - \text{Conf}(B_m) \right|$.

### 2.2 ENSEMBLES

Aggregating the predictions of multiple models into an ensemble is a well-established strategy to improve generalization (Hansen and Salamon, 1990; Perrone and Cooper, 1992; Dietterich, 2000).

**BatchEnsemble:** BatchEnsemble takes a network architecture and shares its parameters across ensemble members, adding only a rank-1 perturbation for each layer in order to decorrelate member predictions (Wen et al., 2020). For a given layer, define the shared weight matrix among $K$ ensemble members as $\mathbf{W} \in \mathbb{R}^{m \times d}$. A tuple of trainable vectors $\mathbf{r}_k \in \mathbb{R}^m$ and $\mathbf{s}_k \in \mathbb{R}^n$ are associated with each ensemble member $k$. The new weight matrix for each ensemble member in BatchEnsemble is

$$\mathbf{W}'_k = \mathbf{W} \circ \mathbf{F}_k, \text{ where } \mathbf{F}_k = \mathbf{r}_k \mathbf{s}_k^\top \in \mathbb{R}^{m \times d}, \qquad (2)$$

where $\circ$ denotes the element-wise product. Applying rank-1 perturbations via $\mathbf{r}$ and $\mathbf{s}$ adds few additional parameters to the overall model. We use an ensemble size of $4$ in all experiments.

**MC-Dropout**: Gal and Ghahramani (2016) interpret Dropout (Srivastava et al., 2014) as an ensemble model, leading to its application for uncertainty estimates by sampling multiple dropout masks at test time in order to ensemble its predictions. We use an ensemble size of $20$ in all experiments.

**Deep Ensembles:** Composing an ensemble of models, each trained with a different random initialization, provides diverse predictions (Fort et al., 2019) which have been shown to outperform strong

baselines on uncertainty estimation tasks (Lakshminarayanan et al., 2017). We use an ensemble size of 4 in all experiments.

In this work, we focus on the interaction between data augmentation strategies and BatchEnsemble, MC-Dropout, and deep ensembles. Other popular ensembling approaches leverage weight averaging such as Polyak-Ruppert (Ruppert, 1988), checkpointing (Huang et al., 2017), and stochastic weight averaging (Izmailov et al., 2018) to collect multiple sets of weights during training and aggregate them to make predictions with only a single set.

## 2.3 DATA AUGMENTATION

Data augmentation encourages a model to make invariant predictions under desired transformations which can greatly improve generalization performance. For example, in computer vision, random left-right flipping and cropping are de-facto approaches (He et al., 2016). We highlight two state-of-the-art techniques which we study.

**Mixup**: Mixup (Zhang et al., 2018) manipulates both the features and the labels in order to encourage linearly interpolating predictions. Given an example $(x_i, y_i)$, Mixup applies

$$\tilde{x}_i = \lambda x_i + (1 - \lambda)x_j, \quad \tilde{y}_i = \lambda y_i + (1 - \lambda)y_j. \tag{3}$$

Here, $x_j$ is sampled from the training dataset (taken from the minibatch), and $\lambda \sim \text{Beta}(a, a)$ for a fixed hyperparameter $a > 0$.

Mixup was shown to be effective for generalization and calibration of deep neural networks (Zhang et al., 2018; Thulasidasan et al., 2019b). Recent work has investigated why Mixup improves generalization (Guo et al., 2018; Shimada et al., 2019) and adversarial robustness (Beckham et al., 2019; Pang et al., 2020; Mangla et al., 2020). Given Mixup's simplicity, many extensions have been proposed with further improvements (Yun et al., 2019; Berthelot et al., 2019; Verma et al., 2019; Roady et al., 2020; Chou et al., 2020).

**AugMix:** Searching or sampling over a set of data augmentation operations can lead to significant improvement on both generalization error and calibration (Cubuk et al., 2019b;a). AugMix (Hendrycks et al., 2020) applies a sum of augmentations, each with random weighting, with a Jensen-Shannon consistency loss to encourage similarity across the augmentations. AugMix achieves state-of-the-art calibration across in- and out-of-distribution tasks. Let $\mathcal{O}$ be the set of data augmentation operations and $k$ be the number of AugMix iterations. AugMix samples $w_1, \ldots, w_k \sim \text{Dirichlet}(a, \ldots, a)$ for a fixed hyperparameter $a > 0$ and $\text{op}_1, \ldots, \text{op}_k$ from $\mathcal{O}$. Given an interpolation parameter $m$, sampled from $\text{Beta}(a, a)$, the augmented input $\tilde{x}_{augmix}$ is:

$$\tilde{x}_{augmix} = m x_{orig} + (1 - m)x_{aug}, \qquad x_{aug} = \sum_{i=1}^{k} w_i \text{op}_i(x_{orig}). \tag{4}$$

## 3 MIXUP-ENSEMBLE PATHOLOGY

We seek to understand the effect of data augmentations on ensembles. In particular, we hope to verify the hypothesis of compounding improvements when combining the seemingly orthogonal techniques of data augmentation and ensembles. To our surprise, we find that augmentation techniques can be detrimental to ensemble calibration.

### 3.1 THE SURPRISING MISCALIBRATION OF ENSEMBLES WITH MIXUP

Ensembles are the most known and simple approaches to improving calibration (Ovadia et al., 2019; Lakshminarayanan et al., 2017), and Thulasidasan et al. (2019b) showed that Mixup improves calibration in a single network. Motivated by this, Fig. 1 applies Mixup to each ensemble member on CIFAR-10/CIFAR-100 with WideResNet 28-10 (Zagoruyko and Komodakis, 2016). Here, we searched over Mixup's optimal hyperparameter $\alpha$ (Eq. 3) and found that $\alpha = 1$ gives the best result, which corroborates the finding in Zhang et al. (2018). All data points in Fig. 1 are averaged over 5 random seeds.

Figs. 1a and 1b demonstrate improved test accuracy (Red (ensembles without Mixup) to Blue (ensembles with Mixup)). However, if we shift focus to Figs. 1c and 1d's calibration error, it is evident that combining Mixup with ensembles leads to *worse* calibration (Red to Blue). This is counter-intuitive as we would expect Mixup, which improves calibration of individual models (Thulasidasan

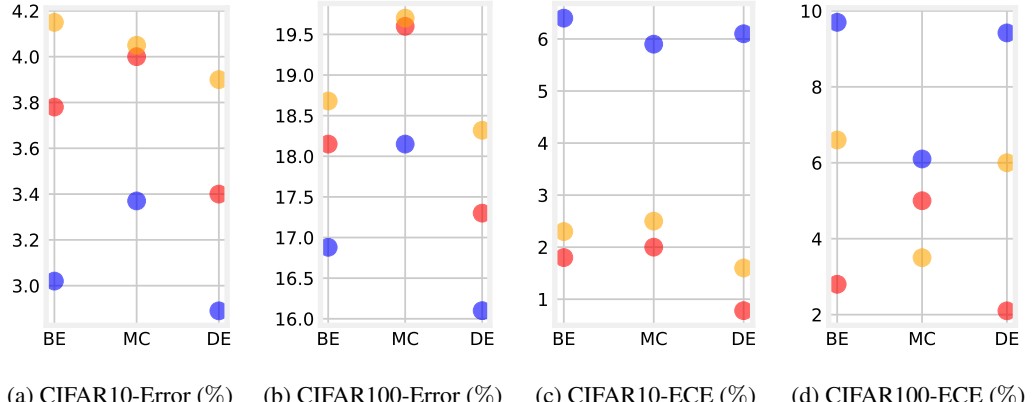

(a) CIFAR10-Error (%)    (b) CIFAR100-Error (%)    (c) CIFAR10-ECE (%)    (d) CIFAR100-ECE (%)

Figure 1: WideResNet 28-10 on CIFAR-10/CIFAR-100. Red: Ensembles without Mixup; Blue: Ensembles with Mixup; Orange: Individual models in ensembles without Mixup. **(a) & (b)**: Applying Mixup to different ensemble methods leads to consistent improvement on test accuracy. **(c) & (d)**: Applying Mixup to different ensemble methods harms calibration. Averaged over 5 random seeds.

et al., 2019a), to also improve the calibration of their ensemble. Fig. 1 confirms this pattern across BatchEnsemble (**BE**), MC-dropout (**MC**), and deep ensembles (**DE**). This pathology also occurs on ImageNet, as seen in Table 1.

**Why do Mixup ensembles degrade calibration?** To investigate this in more detail, Fig. 2 plots a variant of reliability diagrams (DeGroot and Fienberg, 1983) on BatchEnsemble. We bin the predictions into $M = 15$ equally spaced intervals based on their confidence (softmax probabilities) and compute the difference between the average confidence and the average accuracy as in Eq. 1 for each bin. Fig. 2 tracks this difference over varying confidence levels. A positive difference (Acc−Conf) implies under-confidence with respect to the true frequencies; negative implies over-confidence; and zero implies perfect calibration.

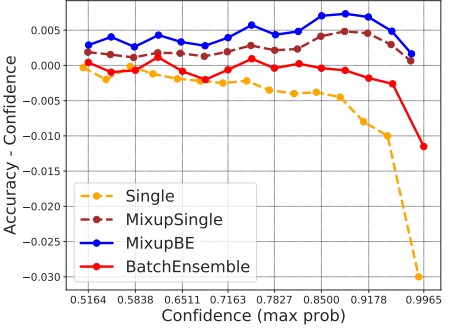

Figure 2: Reliability diagrams on CIFAR-100 with a WideResNet 28-10.

The backbone model in Fig. 2 is BatchEnsemble with an ensemble size of 4 (we also found this consistent for MC-Dropout and Deep-Ensemble). The figure presents 4 methods: Single: vanilla WideResNet 28-10; MixupSingle: WideResNet 28-10 model trained with Mixup; BatchEnsemble: vanilla BatchEnsemble WideResNet 28-10 model; MixupBE: BatchEnsemble WideResNet 28-10 model trained with Mixup. Fig. 2 shows that only models trained with Mixup have positive (Acc − Conf) values on the test set, which suggests that Mixup encourages under-confidence. Mixup ensemble's under-confidence is also greater in magnitude than that of the individual Mixup models. This suggests that Mixup ensembles suffer from compounding under-confidence, leading to a worse calibration for the ensemble than the individual Mixup models' calibration. This is contrary to our intuition that ensembles always improves calibration.

To further visualize this issue, Appendix C's Fig. 8 investigates the confidence (softmax probabilities) surface of deep ensembles and Mixup when trained on a toy dataset consisting of 5 clusters, each with a different radius. We ensemble over 4 independently trained copies of 3-layer MLPs. Deep ensemble's predictive confidence is plotted over the entire input data space in Fig. 8c. The resulting predictions are extremely confident except at the decision boundaries. Deep Ensemble still displays high confidence in the area nearest to the origin which is expected to have lower confidence level. On the other hand, Fig. 8d shows that Mixup-Ensemble is only confident in a very constrained area around the training clusters, leading to an overall under-confident classifier which confirms our postulation of compounding under-confidence.

### 3.2 IS THE PATHOLOGY SPECIFIC TO MIXUP?

At the core of the issue is that Mixup conflates data uncertainty (uncertainty inherent to the data generating process) with model uncertainty. Soft labels can correct for over-confidence in single models which have no other recourse to improve uncertainty estimates. However, when combined with ensembles, which incorporate model uncertainty, this correction may be unnecessary. Because image classification benchmarks tend to be deterministic, soft labels encourage predictions on training data to be less confident about their true targets even if they are correct. We validate this hypothesis by showing it also applies to label smoothing.

**Label Smoothing**: Like Mixup, label smoothing applies soft labels: it smoothens decision boundaries by multiplying a data point's true class by $(1 - \alpha)$, with probability $\alpha$ spread equally across other classes. Using the same experimental setup as before, we apply increasing levels of label smoothing to ensembles of WideResNet 28-10 models trained on CIFAR-10. Fig. 3 demonstrates the harmful effect of label smoothing on CIFAR-10 ECE, particularly when aggressive (coeff $\geq 0.2$). In the concurrent work, Qin et al. (2020) found that label smoothing plus ensemble leads to worse calibration. They showed that adjusting model confidence successfully corrects the compounding underconfidence.

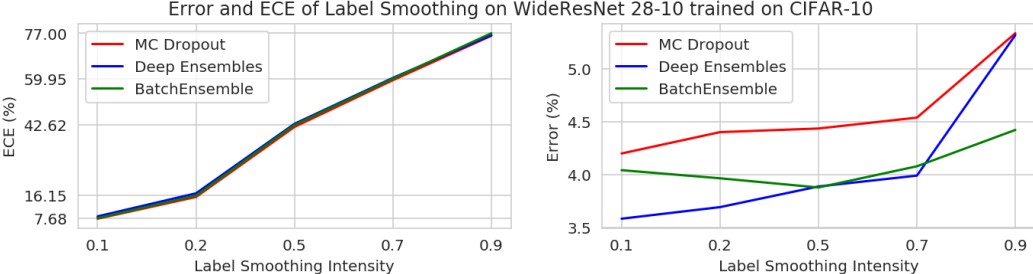

Figure 3: ECE and Error on CIFAR-10 with label smoothing on MC Dropout, Deep Ensembles, and BatchEnsemble. ECE degrades with label smoothing, particularly when it is more aggressive ($\geq 0.2$).

## 4 CONFIDENCE ADJUSTED MIXUP ENSEMBLES (CAMIXUP)

In this section, we aim to fix the compounding under-confidence issue when combining Mixup and ensembles without sacrificing its improved accuracy on both in- and out-of-distribution data.

### 4.1 CLASS BASED CAMIXUP

Mixup encourages model under-confidence as shown in Fig. 2. Notice that Mixup assigns a uniform hyperparameter $\alpha$ to all examples in the training set. To improve Mixup, we start from the intuition that in classification, some classes are prone to be more difficult than others to predict. This can be confirmed by Fig. 4a, which provides examples of per-class test accuracy. Ideally, we prefer our model to be confident when it is predicting over easy classes such as cars and ships. For harder classes like cats and dogs, the model is encouraged to be less confident to achieve better calibration.

Therefore, instead of a uniform Mixup hyperparameter for all classes, we propose to adjust the Mixup hyperparameter of each class by the difference between its accuracy and confidence. CAMixup's intuition is that we want to apply Mixup on hard classes on which models tend to be over-confident. On easy examples, we impose the standard data-augmentation without Mixup. This partially prevents Mixup models from being over-confident on difficult classes while maintaining its good calibration on out-of-distribution inputs.[2]

Denote the accuracy and confidence of class $i$ as $\text{Acc}(C_i)$ and $\text{Conf}(C_i)$. We adjust Mixup's $\lambda$ in Eqn. 3 by the sign of $\text{Acc}(C_i) - \text{Conf}(C_i)$, which are defined as $\text{Acc}(C_i) = \frac{1}{|C_i|} \sum_{x_j \in C_i} \mathbb{1}(\hat{y}_j = i)$ and $\text{Conf}(C_i) = \frac{1}{|C_i|} \sum_{x_j \in C_i} \hat{p}_i$.

$$\lambda_i = \begin{cases} 0 & \text{Acc}(C_i) > \text{Conf}(C_i) \\ \lambda & \text{Acc}(C_i) \leq \text{Conf}(C_i). \end{cases} \quad (5)$$

---

[2] We focus on classification, where classes form a natural grouping of easy to hard examples. However, the same idea can be used on metadata that we'd like to balance uncertainty estimates, e.g., gender and age groups.

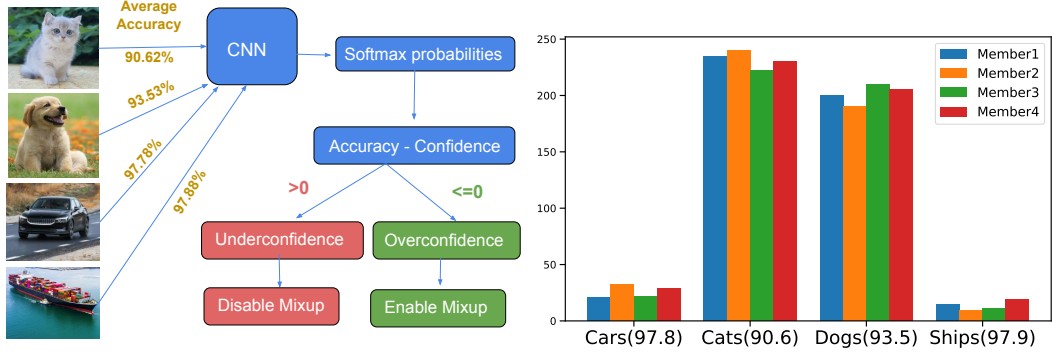

(a) Proposed CAMixup method.

(b) # epochs in which Mixup was applied to each class.

Figure 4: *Left*: An illustration of the proposed CAMixup data augmentation. Selected per-class test accuracies are showed in brown. Overall test accuracy is 96.2% on CIFAR-10; *Right*: Number of epochs (out of 250) where CAMixup enables Mixup for selected classes in BatchEnsemble. CAMixup tends to assign Mixup to hard classes. Counts are accumulated individually for each ensemble member (ensemble size 4).

If the model is already under-confident on class $i$ ($\mathrm{Acc}(C_i) > \mathrm{Conf}(C_i)$), Mixup is not applied to examples in the class, and $\lambda_i = 0$. However, if $\mathrm{Acc}(C_i) \leq \mathrm{Conf}(C_i)$, the model is over-confident on this class, and Mixup is applied to reduce model confidence. We compute the accuracy and confidence on a validation dataset after each training epoch.

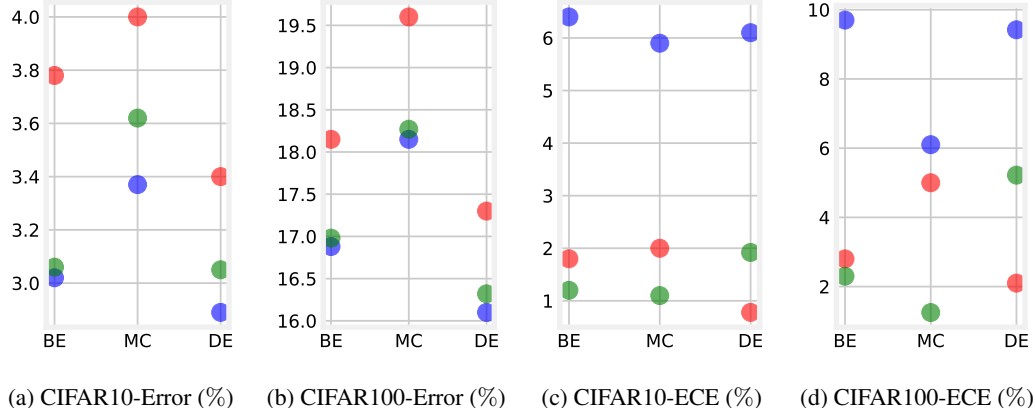

(a) CIFAR10-Error (%)    (b) CIFAR100-Error (%)    (c) CIFAR10-ECE (%)    (d) CIFAR100-ECE (%)

Figure 5: WideResNet 28-10 on CIFAR-10/CIFAR-100. Red: Ensembles without Mixup; Blue: Ensembles with Mixup; Green: Our proposed CAMixup improves both accuracy & ECE of ensembles.

Notice that $\lambda_i$ is dynamically updated at the end of each epoch. To understand which classes are more often assigned Mixup operation, Fig. 4 calculates the number of times that $\lambda_i > 0$ throughout training. The maximum number of times is the number of total training epochs, which is 250 in the BatchEnsemble model. We find that CAMixup rarely enables Mixup to easy classes such as cars and ships: the number of times is less than 10% of the total epochs. For harder classes like cats and dogs, CAMixup assigns Mixup operation almost every epoch, accounting for more than 80% of total epochs. In summary, Fig. 4 shows that CAMixup reduces model confidence on difficult classes and encourages model confidence on easy classes, leading to better overall calibration. Appendix D.1's Fig. 9a also shows that CAMixup effectively shifts the confidence to the lower region.

Fig. 5 presents results of CAMixup on CIFAR-10 and CIFAR-100 test set, where we compare the effect of Mixup and CAMixup on different ensembling strategies (BatchEnsemble, MC Dropout, DeepEnsemble). Adding Mixup to ensembles improves accuracy but worsens ECE. Adding CAMixup to ensembles significantly improves accuracy of ensembles in all cases. More importantly, the calibration results in Figs. 5c and 5d show that CAMixup ensembles are significantly better calibrated

than Mixup ensembles, for instance, CAMixup reduces ECE by more than 5X for BatchEnsemble over Mixup. We observe a minor decrease in test accuracy (at most $0.2\%$) when comparing CAMixup ensembles with Mixup ensembles, but we believe that this is a worthwhile trade-off given the significant improvement in test ECE.

Table 1 presents similar experiments applied to ResNet-50 on ImageNet, using BatchEnsemble as the base ensembling strategy. These results are state of the art to the best of our knowledge: Dusenberry et al. (2020) report 1.7% ECE with Rank-1 Bayesian neural nets and 3.0% with Deep Ensembles; Thulasidasan et al. (2019a) report 3.2% for ResNet-50 with Mixup, 2.9% for ResNet-50 with an entropy-regularized loss, and 1.8% for ResNet-50 with label smoothing.

Table 1: BatchEnsemble with ensemble size 4 on ImageNet.

|  | ACC | ECE |
| --- | --- | --- |
| BatchEnsemble | 77.0 | 2.0% |
| MixupBE | 77.5 | 2.1% |
| CAMixupBE | 77.4 | 1.5% |

### 4.2 PERFORMANCE UNDER DISTRIBUTION SHIFT

Here, we assess model resilience to covariate shift by evaluating on the CIFAR-10-C and CIFAR-100-C benchmarks (C stands for corruptions) proposed by Hendrycks and Dietterich (2019a), which apply 15 types of corruptions each with 5 levels of intensity. We evaluate the performance of CAMixup vs Mixup when applied to different ensembles, and report average error on ECE across different types of corruptions and intensities.

Fig. 6a shows that Mixup improves accuracy on the corrupted dataset because of its strong regularization effect. However, the models tend to be over-confident as one moves further from the original distribution (higher corruption intensities), so encouraging under-confidence is not an issue. This explains why Mixup ensembles maintain low ECE on out-of-distribution test data in Fig. 6b.

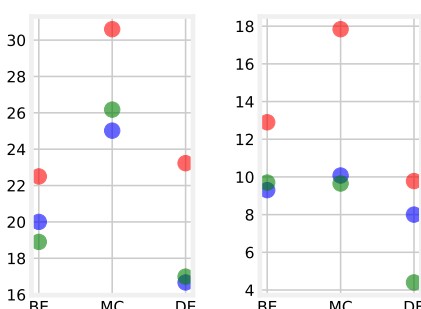

(a) CIFAR10-C-Error    (b) CIFAR10-C-ECE

Figure 6: WideResNet 28-10 on CIFAR-10-C. Red: Ensembles without Mixup; Blue: Ensembles with Mixup; Green: Ensembles with CAMixup (ours).

Fig. 6b also shows that CAMixup's calibration on out-of-distribution data (CIFAR-10-C) is also on par with Mixup ensembles. We observe the same result on CIFAR-100-C (Appendix D.1's Fig. 9). Thus, we successfully improve model calibration on in-distribution datasets without sacrificing its calibration on out-of-distribution datasets.

## 5 COMPOUNDING THE BENEFITS OF CAMIXUP WITH AUGMIX ENSEMBLES

We have investigated why certain data augmentation schemes may not provide complementary benefits to ensembling. We proposed class-adjusted Mixup (CAMixup) which compounds both accuracy and ECE over vanilla ensembles. We believe that the insights from our work will allow the community and practitioners to compound SOTA performance. We provide two concrete examples.

### 5.1 AUGMIX

We show how CAMixup can compound performance over ensembles of models trained with AugMix, which were shown by Hendrycks et al. (2020) to achieve state-of-the-art accuracy and calibration on both clean and corrupted benchmarks. We primarily focus on improving BatchEnsemble and we investigate if adding better data augmentation schemes closes the gap between memory-efficient ensembles (BatchEnsemble) and independent deep ensembles.

As discussed in Section 2.3, AugMix only uses label-preserving transformations. Therefore AugMix provides complementary benefits to ensembles (and CAMixup). This is consistent with calibration improvements in the literature with ensemble methods, which apply standard data augmentation such as random flips, which also do not smoothen labels.

We consider a combination of AugMix and Mixup as it allows the model to encounter both diverse label-preserving augmentations and soft labels under a linearly interpolating regime. The combination

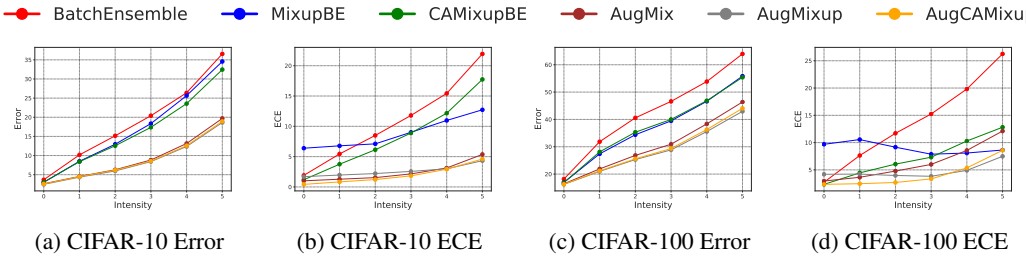

Figure 7: Performance on BatchEnsemble under dataset shift. Mixup and AugMixup improve accuracy and calibration under shift but significantly worsen in-distribution calibration. Our proposed CAMixup and AugCAMixup improve accuracy and calibration.

| Method/Metric | CIFAR-10 | | | CIFAR-100 | | |
|---|---|---|---|---|---|---|
| | Acc(↑) | ECE(↓) | cA/cECE | Acc(↑) | ECE(↓) | cA/cECE |
| AugMix BE | 97.36 | 1.02% | 89.49/2.6% | 83.57 | 2.96% | 67.12/7.1% |
| AugMixup BE | **97.52** | 1.71% | **90.05**/2.8% | **83.77** | 4.19% | **69.26**/4.8% |
| AugCAMixup BE | 97.47 | **0.45%** | 89.81/**2.4%** | 83.74 | **2.35%** | 68.71/**4.4%** |

Table 2: Results for Wide ResNet-28-10 BatchEnsemble on in- and out-of-distribution CIFAR-10/100 with various data augmentations, averaged over 3 seeds. **AugMix**: AugMix + BatchEnsemble; **AugMixup**: AugMix + Mixup BatchEnsemble; **AugCAMixup**: AugMix + CAMixup BatchEnsemble. Adding Mixup to AugMix model increases test accuracy and corrupt accuracy at the cost of calibration decay on testset. CAMixup bridges this gap with only a minor drop in accuracy.

AugMixup (AugMix + Mixup) can be written as

$$x = \lambda * \text{AugMix}(x_1) + (1 - \lambda)\,\text{AugMix}(x_2), \quad y = \lambda * y_1 + (1 - \lambda) * y_2. \tag{6}$$

Consistent with earlier results on Mixup, Table 2 shows combining AugMixup with BatchEnsemble improves accuracy but worsens ECE, leading to under-confidence on in-distribution data. (Appendix D.2's Fig. 10). With our proposed fix CAMixup, the combination AugCAMixup (AugMix + CAMixup) improves calibration while retaining the highest accuracy for ensembles. Fig. 7 shows detailed results on CIFAR-10-C and CIFAR-100-C. Similar to Mixup, AugMixup improves calibration under shift but worsens calibration on in-distribution. However, our proposed AugCAMixup improves accuracy and calibration of ensembles on both clean and corrupted data.

To the best of our knowledge, these results are state-of-the-art in the literature: Dusenberry et al. (2020) report 0.8% ECE and 1.8% ECE for CIFAR-10 and CIFAR-100 along with 8% and 11.7% ECE for corruptions; Guo et al. (2017) report 0.54% and 2.3% ECE for the smaller Wide ResNet 32 on CIFAR-10 and CIFAR-100 with temperature scaling (93% and 72% accuracy), and Ovadia et al. (2019) demonstrated that temperature scaling does not extend to distribution shift.

## 5.2 TEMPERATURE SCALING

In concurrent work, Rahaman and Thiery (2020) consider the interplay between data augmentation and ensembling on calibration. They also find that Mixup ensembles can be under-confident, and propose temperature scaling as a solution. Their core contribution is the same but differ in slight ways: we further this analysis by showing the compounding under-confidence extends to other techniques applying soft labels such as label smoothing, and we propose CAMixup as a solution. Post-hoc calibration techniques like temperature scaling are complementary to our proposal and do not address the core conflation issue with Mixup. Corroborating findings of Ovadia et al. (2019), Appendix G shows combining CAMixup and temperature scaling can further improve test calibration error; it does not improve out-of-distribution calibration. Another concurrent work showed that calibrated ensemble members do not always lead to calibrated ensemble predictions (Anonymous, 2021).

## 6 CONCLUSION

Contrary to existing wisdom in the literature, we find that combining ensembles and Mixup consistently degrades calibration performance across three ensembling techniques. From a detailed

analysis, we identify a compounding under-confidence, where Mixup's soft labels (and more broadly, label-based augmentation strategies) introduce a negative confidence bias that hinders its combination with ensembles. To correct this, we propose CAMixup, which applies Mixup to only those classes on which the model tends to be over-confident, modulated throughout training. CAMixup combines well with state-of-the-art methods. It produces new *state-of-the-art calibration* across CIFAR-10, CIFAR-100, and ImageNet while obtaining competitive accuracy. Appendix H points out potential future work and limitations of CAMixup.

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

## A  DATASET DETAILS

**CIFAR & CIFAR-C:** We consider two CIFAR datasets, CIFAR-10 and CIFAR-100 (Krizhevsky, 2009). Each consists of a training set of size 50K and a test set of size 10K. They are natural images with 32x32 pixels. Each class has 5,000 training images and 500 training images on CIFAR-10 and CIFAR-100 respectively. In our experiments, we follow the standard data pre-processing schemes including zero-padding with 4 pixels on each sise, random crop and horizon flip (Romero et al., 2015; Huang et al., 2016; Srivastava et al., 2015). If a training method requires validation dataset such as CAMixup, we use separate 2,500 images from 50K training images as the validation set.

It's important to test whether models are well calibrated under distribution shift. CIFAR-10 corruption dataset (Hendrycks and Dietterich, 2019a) is designed to accomplish this. The dataset consists of 15 types of corruptions to the images. Each corruption types have 5 intensities. Thus, in total CIFAR-10C has 75 corrupted datasets. Notice that the corrupted dataset is used as a testset without training on it. Ovadia et al. (2019) benchmarked a number of methods on CIFAR-10 corruption. Similarly, we can apply the same corruptions to CIFAR-100 dataset to obtain CIFAR-100C.

**ImageNet & ImageNet-C:** We used the ILSVRC 2012 classification dataset (Deng et al., 2009) which consists of a total of 1.2 million training images, 50,000 validation images and 150,000 testing images. Images span over 1,000 classes. We follow the data augmentation scheme in He et al. (2016), such as random crop and random flip, to preprocess the training images. During testing time, we apply a 224x224 center crop to images. Similarly to CIFAR-C, we apply 15 corruption types with 5 intensities each to obtain ImageNet-C (Hendrycks and Dietterich, 2019b).

## B  HYPERPARAMETERS IN SECTION 3

We kept the same set of hyperparameters as the BatchEnsemble model in Wen et al. (2020). All hyperparameters can be found in Table 3. The most sensitive hyperparameter we found is whether to use ensemble batch norm, which applies a separate batch norm layer for each ensemble member; and the value of random_sign_init, which controls the standard deviation of Gaussian distributed initialization of $s$ and $r$. We kept BatchEnsemble CIFAR-10 the same as Wen et al. (2020), which does not deploy ensemble batch norm. We enable ensemble batch norm on CIFAR-100 and ImageNet. This allows us to use larger standard deviation in the initialization. The random_sign_init is $-0.5$ on CIFAR-10 and $-0.75$ on CIFAR-100 and -0.75 on ImageNet. In the code, we use negative value to denote the standard deviation of Gaussian distribution (positive value instead initializes with a Bernoulli distribution under that probability). In our case, we only use negative random_sign_init, which means we only consider Gaussian distributed initialization in this work.

| **Dataset** | CIFAR-10 | CIFAR-100 |
|---|---|---|
| ensemble_size | 4 | |
| base_learning_rate | 0.1 | |
| per_core_batch_size | 64 | |
| num_cores | 8 | |
| lr_decay_ratio | 0.1 | |
| train_epochs | 250 | |
| lr_decay_epochs | [80, 160, 200] | |
| l2 | 0.0001 | 0.0003 |
| random_sign_init | 0.5 | 0.75 |
| SyncEnsemble_BN | False | True |

Table 3: Hyperparameters we used in Section 3 regarding to BatchEnsemble. The difference between CIFAR-10 and CIFAR-100 is l2, random_sign_init and whether to use SyncEnsemble_BN.

## C  EXCESSIVE UNDER-CONFIDENCE ON SYNTHETIC DATA

To further understand the confidence surface of Mixup + Ensembles, we provided a visualization in Fig. 8. We trained on a synthetic dataset consisting of 5 clusters, each with a different radius. We ensemble over 4 independently trained copies of 3-layer MLPs. We plotted the softmax probabilities surface of Mixup-Single model, Deep-Ensemble and Mixup-Ensemble. The softmax probabilities

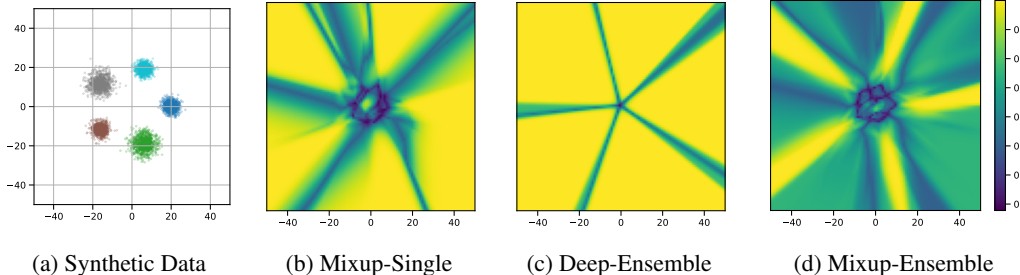

| (a) Synthetic Data | (b) Mixup-Single | (c) Deep-Ensemble | (d) Mixup-Ensemble |

Figure 8: Softmax probabilities surface of different ensemble methods (ensemble size 4) in the input space after training on synthetic data. Deep ensemble is over-confident in the area around origin. Mixup-Ensemble leads to gloabl under-confidence.

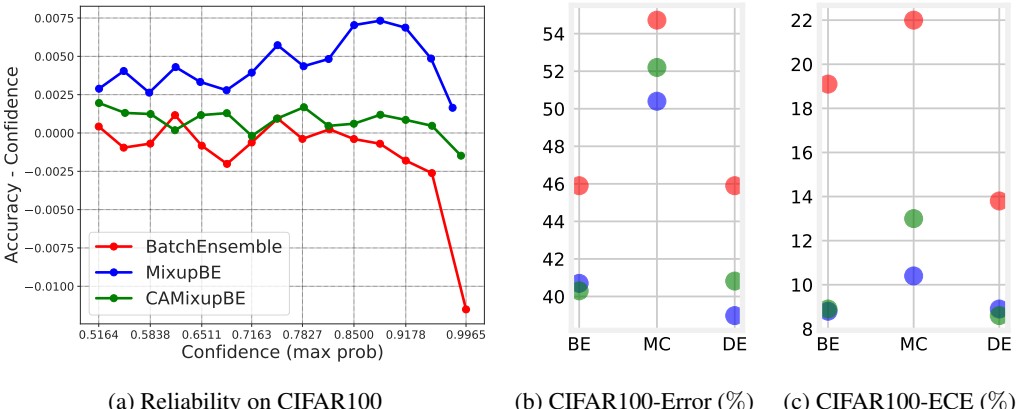

| (a) Reliability on CIFAR100 | (b) CIFAR100-Error (%) (c) CIFAR100-ECE (%) |

Figure 9: *Left*: Reliability diagrams on CIFAR-100 with a WideResNet 28-10. Our proposed CAMixup successfully fixes the under-confidence of Mixup BatchEnsemble, leading to better calibration. **(b) & (c)**: Red: Ensembles without Mixup; Blue: Ensembles with Mixup; Green: Our proposed CAMixup does not harm the out-of-distribution performance.

represent the model confidence. Fig. 8c shows that Deep-Ensemble predictions are extremely confident except at the decision boundaries. Fig. 8b displays a lower confidence than Deep-Ensemble. This is beneficial in the single model context because single deep neural networks tend to be over-confident and Mixup can partially correct this bias. On the other hand, Fig. 8d shows that Mixup-Ensemble is only confident in a very constrained area around the training clusters, leading to an overall under-confident classifier which confirms our postulation of compounding under-confidence.

## D   MORE CALIBRATION RESULTS OF MIXUP-BATCHENSEMBLE

In Section 3.1, we demonstrated that combining Mixup and ensembles leads to worse calibration on testset. In this appendix section, we complement the above conclusion with the analysis on corrupted datasets and with data-augmentation techniques like AugMix.

### D.1   SUPPLEMENTARY RESULTS ON CAMIXUP

In this section, we provided supplementary results on CAMixup. Fig. 2 shows that combining Mixup and BatchEnsemble leads to excessive under-confidence. In Fig. 9a, we showed that our proposed CAMixup fixes this issue by correcting the confidence bias. This explains why CAMixup achieves better calibration on in-distribution testset. As demonstrated in Section 4.2, Mixup improves model out-of-distribution performance because of its strong regularization effect. We showed that our proposed CAMixup inherits Mixup's improvement on CIFAR-10-C. Fig. 9b and Fig. 9c show that this conclusion seamlessly transfers to CIFAR-100-C. We also supplement Fig. 5 with Table 4 and Table 5, illusrating detailed numbers.

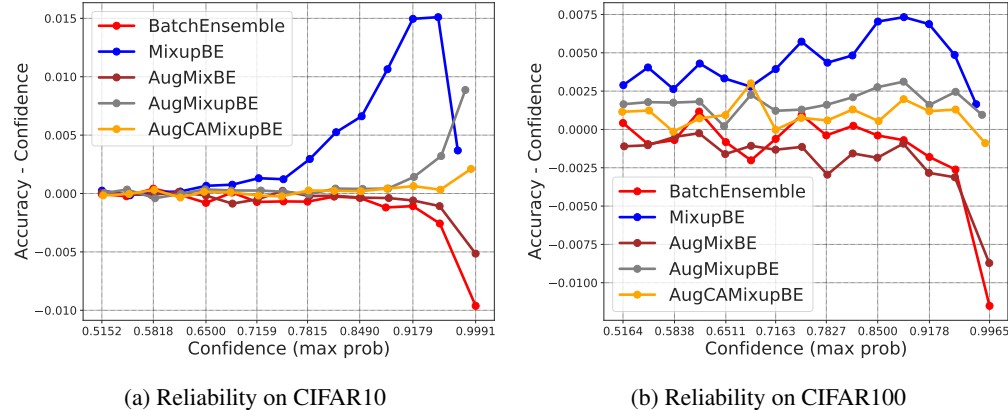

(a) Reliability on CIFAR10        (b) Reliability on CIFAR100

Figure 10: Reliability diagrams on CIFAR-10 and CIFAR-100. Both plots show that AugMix does not lead to under-confidence when combined with ensembles. However, if we combine AugMix with Mixup (AugMixup), the compounding under-confidence issue still exists, leading to suboptimal calibration. Our proposed AugCAMixup corrects this underconfidence bias.

## D.2 SUPPLEMENTARY RESULTS ON AUGMIX

We show that Mixup does not combine with ensembles without sacrificing in-distribution calibration in Section 3.1. As discussed in Section 2.3, AugMix only uses label-preserving transformations and does not modify the labels. Intuitively, it does not reduce model confidence. We support this intuition with Fig. 10. It shows that AugMix does not lead to under-confidence. Therefore it can be combined with ensembles without any calibration issue.

In Table 2, we showed that combining AugMix and Mixup leads to worse calibration due to the under-confidence although AugMix itself does not. To better understand the insights beyond staring at scalars, we provided the reliability diagram analysis as well. In Figure 10, we showed that the under-confidence issue of AugMixup (Augmix + Mixup) still exists. It suggests that applying CAMixup to Augmix can correct the under-confidence bias as what we showed in Fig. 10a and Fig. 10b. Our proposed CAMixup allows to compound performance of ensembles and data augmentation to achieve the best possible performance.

| Method/Metric | CIFAR-10 | | |
| --- | --- | --- | --- |
| | Acc($\uparrow$) | ECE($\downarrow$) | cA/cE |
| BatchEnsemble | 96.22 $\pm$0.07 | 1.8 $\pm$0.2 % | 77.5$\pm$0.3 /12.9 $\pm$1.2 % |
| Mixup BE | **96.98**$\pm$0.08 | 6.4 $\pm$0.4 % | 80.0$\pm$0.4 /**9.3**$\pm$0.3 % |
| CAMixup BE | 96.94 $\pm$0.10 | **1.2** $\pm$0.2 % | **81.1**$\pm$0.4 /9.7$\pm$0.35% |

Table 4: CIFAR-10 results for Wide ResNet-28-10 BatchEnsemble (Wen et al., 2020) (**BE**), averaged over 5 seeds. This table is used to supplement Fig. 5.

| Method/Metric | CIFAR-100 | | |
| --- | --- | --- | --- |
| | Acc($\uparrow$) | ECE($\downarrow$) | cA/cE |
| BatchEnsemble | 81.85$\pm$0.09 | 2.8$\pm$0.1% | 54.1$\pm$0.3/19.1$\pm$0.8% |
| Mixup BE | **83.12**$\pm$0.08 | 9.7$\pm$0.5 % | 59.3$\pm$0.3/**8.8**$\pm$0.4% |
| CAMixup BE | 83.02$\pm$0.10 | **2.3**$\pm$0.1% | **59.7**$\pm$0.3/8.9$\pm$0.4% |

Table 5: CIFAR-100 results for Wide ResNet-28-10 BatchEnsemble (Wen et al., 2020) (**BE**), averaged over 5 seeds. This table is used to supplement Fig. 5.

| Dataset | CIFAR-10 | | | CIFAR-100 | | |
|---|---|---|---|---|---|---|
| Metric | Acc(↑) | ECE(↓) | cA/cE | Acc(↑) | ECE(↓) | cA/cE |
| Deep Ensembles | 96.66 | 0.78% | 76.80/9.8% | 82.7 | **2.1%** | 54.1/13.8% |
| Mixup DE | 97.11 | 6.15% | 83.33/8.0% | 83.90 | 9.42% | 61.02/8.9% |
| CAMixup DE | 96.95 | 1.92% | 83.01/4.4% | 83.68 | 5.22% | 59.18/8.6% |
| AugMix DE | 97.39 | **0.59%** | 89.50/**3.3%** | 84.15 | 5.13% | 68.21/6.7% |
| AugMixup DE | **97.56** | 2.71% | **90.03**/4.3% | **84.85** | 6.86% | **69.31**/7.6% |
| AugCAMixup DE | 97.48 | 1.89% | 89.94/4.7% | 84.64 | 5.29% | 69.19/**5.9%** |

Table 6: Mixup/AugMix/AugMixup/AugCAMixup on **deep ensembles**. We can conclude that Mixup worsens ensemble predictions in deep ensembles as well as in BatchEnsemble. This suggests we can use CAMixup on deep ensembles as well. However, the improvement is not as obvious as it is on BatchEnsemble, leading to the fact that AugMix is the most calibrated (in- and out-of-distribution) data augmentation strategy on deep ensembles.

## E  DEEP ENSEMBLES WITH MIXUP

We showed that CAMixup improves Mixup BatchEnsemble calibration on testset without undermining its calibration under distribution shift in Section 4. In this section, we show that the improvement can also be observed on deep ensembles. In Fig. 11, we showed the under-confidence bias we observed on Mixup + BatchEnsemble also exists on Mixup + deep ensembles, with an even more obvious trend. Beyond commonly used ECE measure, we also explore other calibration measures. They further confirmed our under-confidence intuition. We provide some brief explanation on how to calculate ACE, SCE and TACE.

ACE measure is the same as ECE except for the binning scheme. Rather than equally divide the confidence evenly into several bins, ACE choses an adaptive scheme which spaces the bin intervals so that each contains an equal number of predictions. SCE is the same as ECE except that it accounts for all classes into calibration measure rather than just looking at the class with maximum probability. The softmax predictions induce infinitesimal probabilities. These tiny predictions can wash out the calibration score. TACE is proposed to set a threshold to only include predictions with large predictive probability, to address the above issue.

We present the results of Mixup, CAMixup, AugMix, AugMixup and AugCAMixup on deep ensembles in Table 6. We notice that the improvement of CAMixup on deep ensembles is smaller than its improvement on BatchEnsemble. We postulate that this is because Mixup + deep ensembles is much badly calibrated than Mixup + BatchEnsemble. For example, AugMixup + deep ensembles achieve 2.71% and 6.86% ECE on CIFAR-10 and CIFAR-100. In the meanwhile, AugMixup + BatchEnsemble achieve 1.71% and 4.19%. Thus, even if CAMixup can improve the calibration of Mixup + deep ensembles, it still cannot beat AugMix + deep ensembles. As a result, when we say we close the calibration gap between BatchEnsemble and deep ensembles, we are comparing AugCAMixup BatchEnsemble (BatchEnsemble + CAMixup + Augmix) to AugMix deep ensembles. This is because AugMix deep ensembles achieve the best calibration among all variants we tried. How to completely fix the under-confidence in deep ensembles is a natural extension of this work. Since we focus on bridging the calibration gap between BatchEnsemble and deep ensembles, we delegate the complete fix in deep ensembles to the future work.

## F  METRICS OTHER THAN ECE

ECE is the standard metric in calibration, but it is a biased estimate of true calibration (Vaicenavicius et al., 2019). Heavily relying on ECE metric might lead to inconsistent conclusion. In this section, we computed the calibration error with recently proposed calibration estimator which reduces bias in ECE, including debiased calibration estimator (Kumar et al., 2019) (DCE) and SKCE (Widmann et al., 2019). fig. 12 shows that our conclusion in the main section are also supported by these two recently proposed calibration estimators. In particular, the improvement of proposed CAMixup over

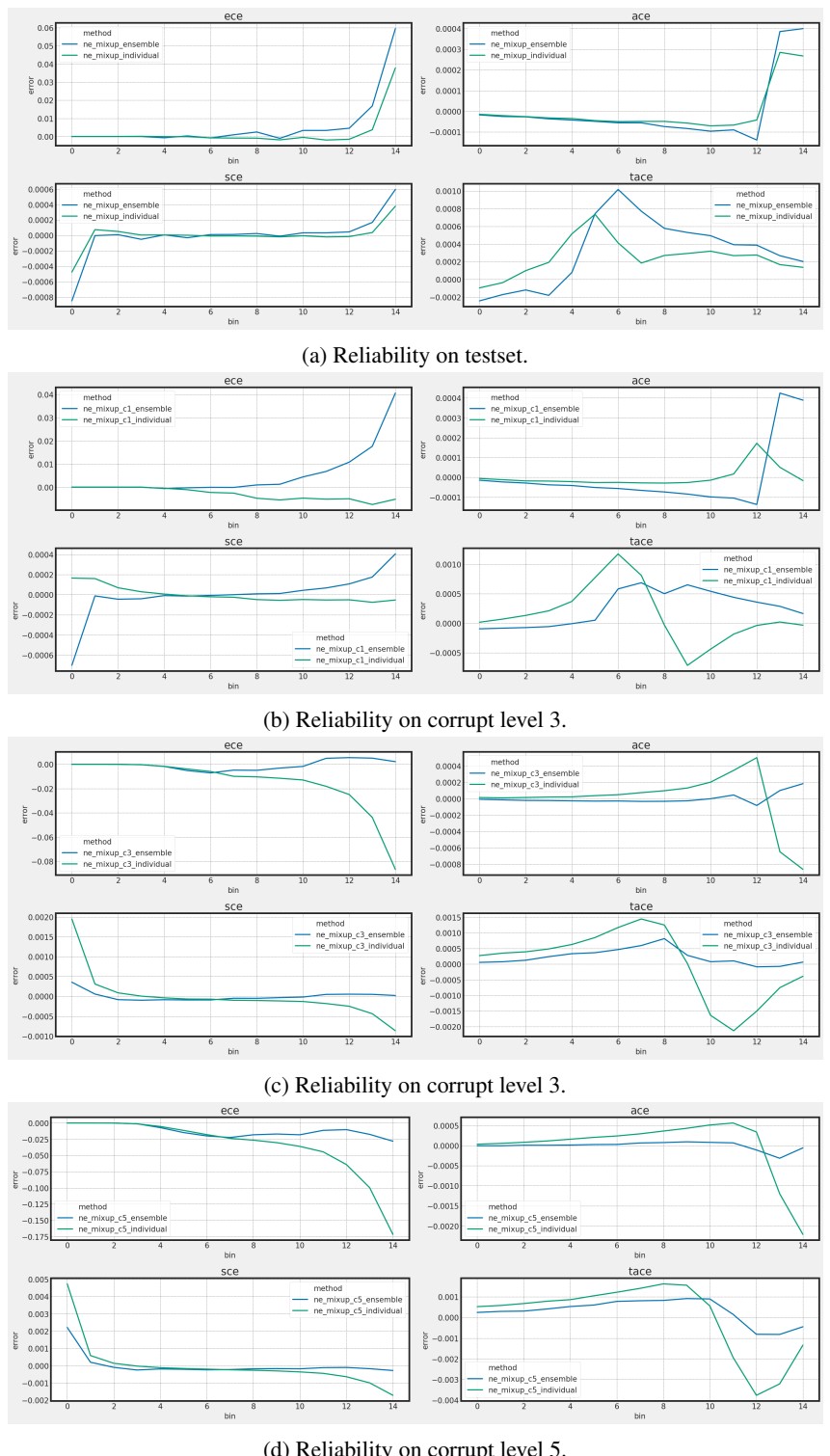

(a) Reliability on testset.

(b) Reliability on corrupt level 3.

(c) Reliability on corrupt level 3.

(d) Reliability on corrupt level 5.

Figure 11: WideResNet-28-10 **Deep Ensembles** with Mixup on CIFAR-10. We plotted the reliability diagram of ensemble and individual predictions. Besides ECE, we also plotted other calibration metrics such as ACE, SCE and TACE proposed in Nixon et al. (2019). All metrics verify the conclusion that Mixup + Ensembles leads to under-confidence on testset.

| Method/Metric | BatchEnsemble | | | Deep-Ensembles | | |
| --- | --- | --- | --- | --- | --- | --- |
| | Acc($\uparrow$) | SKCE($\downarrow$) | cA/cSKCE | Acc($\uparrow$) | SKCE($\downarrow$) | cA/cSKCE |
| Vanilla | 96.22 | 3.4e$-$4 | 77.5/0.026 | 96.66 | **3.4**e$-$5 | 54.1/0.018 |
| Mixup | **96.98** | 4e$-$3 | 80.0/0.024 | **97.11** | 4.4e$-$3 | 59.3/0.0068 |
| CAMixup | 96.94 | **1.3**e$-$4 | **81.1/0.019** | 96.95 | 4.3e$-$4 | **59.7/0.0032** |

Table 7: Results for Wide ResNet-28-10 BatchEnsemble (Wen et al., 2020) and Deep Ensembles on CIFAR-10 and CIFAR-10-C, averaged over 3 seeds. This table is used to supplement Fig. 12

Mixup on testset is even larger than what ECE reflects in Fig. 5. Table 7 demonstrates the specific numbers used in Fig. 12.

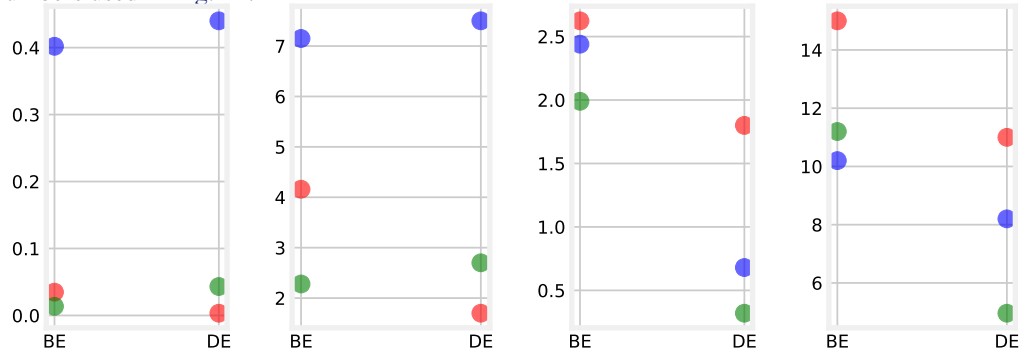

(a) CIFAR10-SKCE (%)    (b) CIFAR10-DCE (%)    (c) CIFAR10-C-SKCE (%)   (d) CIFAR10-C-DCE (%)

Figure 12: WideResNet 28-10 on CIFAR-10 and CIFAR-10-C, averaged over 3 random seeds. **SKCE**: Squared kernel calibration error computed in Widmann et al. (2019). **DCE**: Debiased calibration error in Kumar et al. (2019). Red: Ensembles without Mixup; Blue: Ensembles with Mixup; Green: Ensembles with CAMixup (ours). Both SKCE and DCE give consistent rankings on calibration error to the ranking in Fig. 5 and Fig. 6. This plot shows that our proposed CAMixup is effective in reducing Mixup calibration error when combined with ensembles.

## G   CAMIXUP WITH TEMPERATURE SCALING

See Fig. 13.

# H   LIMITATIONS AND FUTURE WORK

We describe limitations of our work, signalling areas for future research. One limitation of CAMixup is that all examples in the same class still share the same Mixup coefficient. This leaves room for developing more fine-grained adaptive Mixup mechanisms, such as adapting the Mixup coefficient per example. This relates to an open research question: how do you measure the training difficulty of a data point given a deep network? (Toneva et al., 2018; Agarwal and Hooker, 2020) Another limitation is we showed that CAMixup still cannot fully fix the miscalibration of Mixup + deep ensembles in Appendix E, due to the fact that Mixup + deep ensembles leads to even worse calibration than Mixup + BatchEnsemble. This raises a harder question which CAMixup cannot completely solve but also leaves more research room to further understand why Mixup is worse on deep ensembles and how to address it. Thus, we leave the question on how to address the above issues to future work. Next, we determine whether to use Mixup based on the reliability (Mean Accuracy - Mean Confidence) of each class on validation set. One concern is that CAMixup does not scale well to a large number of classes. Fortunately, we showed that this works on problems up to 1000 classes (ImageNet). Additionally, Mixup has been most successful in the vision domain, hence our focus; and with preliminary success on tabular data and natural language processing (Zhang et al., 2018; Guo et al., 2019). Assessing whether CAMixup and ensembling techniques translate to text is an interesting area.

We took a first step in developing a more fine-grained adaptive Mixup mechanism. Recall that class based CAMixup calculates the reliability (Accuracy - Confidence) at the end of each epoch, then it decided whether to apply Mixup in each class (illustrated in Fig. 4). This requires extra computation on validation dataset and it assigns uniform Mixup coefficient within one class. By leveraging recently developed forgetting count (Toneva et al., 2018), we can adjust Mixup coefficient for each example based on its forgetting counts. The intuition is if an examples is associated with high forgetting counts, it indicates the model tends to forget this example. To achieve better calibration, we should place low confidence on this example. The algorithm of forgetting counts based CAMixup is presented in Algorithm 1. In summary, we first calculate the forgetting counts for each training example and obtain the median of these counts as the threshold. Then, CAMixup applies Mixup to the training example whose forgetting counts are higher than the median.

**Algorithm 1** Forgetting Count Based CAMixup

---

initialize $\text{prevacc}_i = 0, i \in D$
initialize $\text{forgetting } T[i] = 0, i \in D$
initialize $\text{MixupCoeff}[i] = 0$
**while** training **do**
    $B \sim D$ # sample a minibatch
    Apply Mixup on $B$ based on MixupCoeff
    **for** $\text{example}_i \in B$ **do**
        compute $\text{acc}_i$
        **if** $\text{prevacc}_i > \text{acc}_i$ **then**
            $T[i] = T[i] + 1$
            $\text{prevacc}_i = \text{acc}_i$
        **end if**
    **end for**
    gradient update classifier on B
    $\text{rank} = sort(T)$
    $\text{threshold} = rank[|D|//2]$
    **for** $\text{example}_i \in B$ **do**
        **if** $T[i] > \text{threshold}$ **then**
            $\text{MixupCoeff}[i] = a$
        **else**
            $\text{MixupCoeff}[i] = 0$
        **end if**
    **end for**
**end while**

---

We provided a preliminary results on CIFAR-10 in Fig. 14. It demonstrates that forgetting counts based CAMixup outperforms class based CAMixup on most metrics across BatchEnsemble and MC-dropout. One exception is that it underperforms on test calibration on MC-dropout. We could not observe the same improvement on CIFAR-100. We postulate that the reliability of forgetting count on CIFAR-100 is not as good as it is on CIFAR-10, leading to the inconsistent results. We leave the question on how to improve forgeting count based CAMixup on CIFAR-100 into future work.

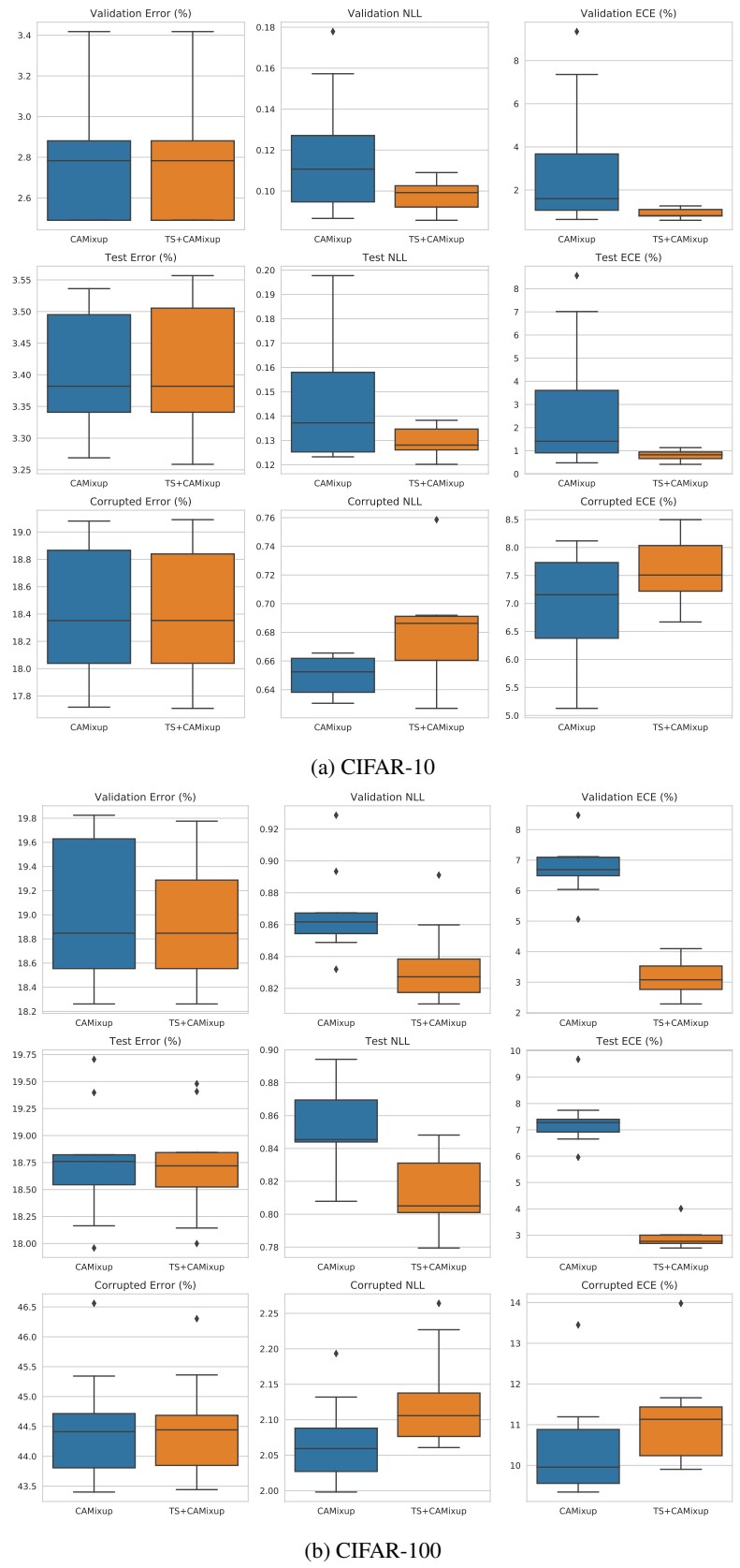

Figure 13: Combining CAMixup and Temperature Scaling further improves test ECE. It does not make further improvements on out-of-distribution calibration however.

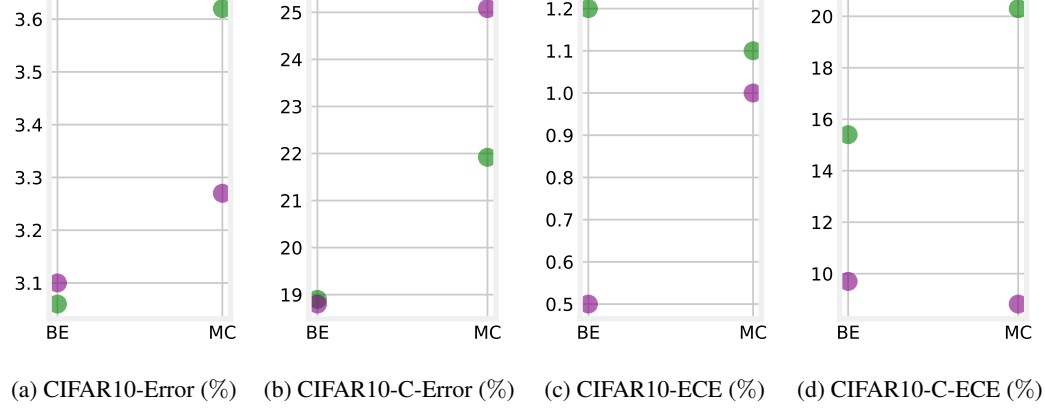

(a) CIFAR10-Error (%)    (b) CIFAR10-C-Error (%)    (c) CIFAR10-ECE (%)    (d) CIFAR10-C-ECE (%)

Figure 14: WideResNet 28-10 on CIFAR-10 and CIFAR-10-C. Green: Class based CAMixup. Purple: Forgetting count based CAMixup. Forgetting count based CAMixup outperforms class based Mixup in most metrics across BatchEnsemble and MC-dropout.

