# OpenReview forum: "Combining Ensembles and Data Augmentation Can Harm Your Calibration"
_ICLR.cc/2021/Conference — ICLR 2021 Poster_

### Official Review · AnonReviewer2 · 2020-10-15
**Review: Combining Ensembles and Data Augmentation Can Harm Your Calibration**

**Rating:** 7
**Confidence:** 5

**Review:**

##########################################################################
Summary:

The paper identifies the negative effects on calibration and the robustness of the deep models when data augmentation and the ensembles are combined. The paper proposes a technique that mitigates this drawback and improves the calibration and robustness across benchmarks.

##########################################################################

Reasons for score:

Overall, I vote for accepting. The observations in the paper and the improvements to the state-of-the-art are certainly very encouraging. My minor concerns are on clarity of the writing, references at some places. Hopefully the authors will address these concerns in the rebuttal period.
##########################################################################


Pros:

+ Overall, the paper is well written, easy to follow and understandable.

+ The paper is positioned in the body of the literature and does contribute to further improve the state-of-the-art.

+ The pathological effect of augmentations combined with ensembles is an interesting observation. The proposed solution does improve the state-of-the-art both for in-distribution and out-of-distribution data.

+ In section 3.2 the label smoothing experiment is interesting.

+ The additional experiments in Appendix and the discussion on limitations and potential for future directions is useful and much appreciated.

Cons:

- My first major concern is omitting the key references when the terms are introduced for the first time or some of the assertions. Some of them are listed below.

- “Ensemble methods are a simple approach to improve a model’s calibration and robustness.” Who found this? This statement requires a citation.

- “For example, the majority of uncertainty models in vision ...” requires a citation.

- “Deep Ensembles” -- need citation, etc. Please fix all such instances

- “and Hendrycks et al. (2020) highlights further improved results in AugMix when combined with Deep Ensembles. However, we find their complementary benefits are not universally true.” From the above sentence, it is not clear what the findings of Hendrycks et al is and for which of their findings the authors in this paper are contradicting with.  I know the paper is about robustness and uncertainty and the authors refer to that. Be clear in the presentation, think that these papers will be read by ML enthusiasts.

- “... poor calibration of combining ensembles and Mixup on CIFAR”, here it is worth to introduce the kind of ensemble(s) that are used in combining.

- DeepEnsembles, BatchEnsemble, MC-Dropout each of them have different ensemble sizes in the experiments. What is the reasoning behind those hyperparameter selections?

- The calibration pathology is observed for ensembles with mixup, is it true for other data augmentations such as rotation, cropping, etc?

- “Ensembles are the among the most known and simple ...” this sentence doesn’t flow well.

---

> ### Author Response · Authors · 2020-11-18
> **We fixed all citations and updated a more reliable metric than ECE as requested by other reviewers.**
>
> We would like to thank the reviewer for the feedback and suggestions to improve this paper. We address the reviewer’s concerns:
>
> We fixed all citations the reviewer mentioned in the revision. In particular, for Deep Ensembles, there is no standard citation for ensembling deep neural networks from random initializations. So we cited some earliest works on neural network ensembles (Hansen and Salamon, 1990; Krogh and Vedelsby, 1995). In terms of the contradiction to Hendrycks et al. (2020)’s finding on combining AugMix with ensembles, we meant to say the complementary benefits between data augmentations and ensembles are not universally true. One conclusion from our paper is that data augmentation which leaves labels unchanged such as AugMix can be combined with ensembles without hurting calibration. Thus, our paper doesn’t contradict Hendrycks et al (2020)’s finding.
>
> Q: DeepEnsembles, BatchEnsemble, MC-Dropout each of them have different ensemble sizes in the experiments. What is the reasoning behind those hyperparameter selections?
>
> Deep ensembles and BatchEnsemble have the same ensemble size 4 (we had a typo in Section 2 which is fixed in the revision). MC-dropout has ensemble size 20 because it needs a relatively larger sample size to be competitive to deep ensembles and BatchEnsemble.
>
> Q: The calibration pathology is observed for ensembles with mixup, is it true for other data augmentations such as rotation, cropping, etc?
>
> The pathology is because mixup generates in-between labels (soft labels) across all data, reducing model confidence universally. This is problematic when combined with ensemble methods (strong label smoothing which doesn’t augment input data also hurts calibration). We postulate that data augmentation which doesn’t touch labels, such as cropping and rotation, will not affect calibration. In fact, AugMix is a stronger version of cropping and rotation and it doesn’t hurt calibration error as shown in the paper.

---

> > ### Comment · AnonReviewer2 · 2020-11-24
> > **Good work**
> >
> > The authors have addressed all the concerns and gave clarifications. However, I stick with my score of 7 mainly because the paper is incremental in nature.

---

### Official Review · AnonReviewer4 · 2020-10-27
**This work provides an interesting analysis of the interaction between data-augmentation strategies such as MixUp and model ensembles with regards to calibration. This work proposes a novel solution to the problem which achieves good results.**

**Rating:** 8
**Confidence:** 4

**Review:**

This work analyses the interaction between data-augmentation strategies such as MixUp and model ensembles with regards to calibration performance. The authors note how strategies such as mixup and label smoothing, which reduce a single model's over-confidence, lead to degradation in calibration performance when such models are combined as an ensemble. Specifically, all techniques, taken individually, improve calibration by reducing overconfidence. However, in combination they lead to under-confident models and, therefore, worse calibration. Based on this analysis, the author's provide a simple technique which yields SOTA calibration performance on CIFAR-10, CIFAR-10-C, CIFAR-100 and CIFAR-100-C and ImageNet. The authors propose to dynamically enable and disable MixUp based on whether the model is over/under confident on a particular class, as judged on a validation dataset.

I think this work provides useful insight and a simple and effective solution. Additionally, it is clearly written and very easy and pleasant to read.

The authors may find this concurrent work on ensemble calibration to be relevant: https://openreview.net/forum?id=wTWLfuDkvKp

The only question is have is why the authors think that models are overconfident on hard examples/classes and properly calibrated on easy ones. My intuition would be the opposite - that models are overconfident in areas where there are too few training examples and/or areas where there is no data uncertainty. If there is no data uncertainty then overconfidence would not be a problem. So mainly the issue is due to data sparsity in areas of non-zero data uncertainty. Would be good to expand the discussion.

---

> ### Author Response · Authors · 2020-11-18
> **We updated  the paper with a more reliable metric than ECE as requested by other reviewers**
>
> We would like to thank the reviewer for the positive feedback.
>
> As you mentioned, models are overconfident in areas where there are too few training examples. Examples in this area are hard examples so models are overconfident on these examples. Intuitively, training difficulty of one class is measured by its validation accuracy in our paper. DNN tends to be overconfident in every class. Namely, the expected confidence in each class is about the same. However, the accuracy is significantly lower in difficult classes. Overconfidence is measured by confidence - accuracy. Therefore, models are more overconfident on hard classes/examples than easy classes/examples.
>
> In the revision (Section 5.2), we also cited the concurrent paper you mentioned in the review.

---

### Official Review · AnonReviewer1 · 2020-10-27

**Rating:** 7
**Confidence:** 4

**Review:**

After the discussion, my concerns were fixed. The paper explores the interesting relations of Mix Up and Uncertainty, which is useful and will be the right fit for the conference.

**Summary:**

The work studies how a better calibration of individual members of an ensemble affects the calibration of an ensemble. It is demonstrated that i) better calibration of individual members of the ensemble may lead to the worse calibration of the ensemble predictions ii) this is the case when mix-up / label smoothing are used during training.

To fix the issue Confidence Adjusted mixup Ensembles (CAMixup) is proposed. The CAMixup is an adaptive mixup data augmentation based on per-class calibration criteria. The core idea of CAMixup is to use powerful (unconfidence encouraging) mixup data-augmentation on examples of overconfident classes, and do not use mixup for the under-confident classes. The confidence criteria are computed once in an epoch.

The empirical results are provided in- and out-of-domain for CIFARs(C) and ImageNet(C).

**The concerns:**

1) ECE is a biased estimate of true calibration with a different bias for each model, so it is not a valid metric to compare even models trained on the same data [Vaicenavicius2019]. In other words, the measured ECE has no guaranty to have something to do with the real calibration but reflects the bias of the measured metric. Other metrics, that are based on histogram estimates have the same problem. Please put extra attention to this concern.

What I suggest is the following:

a. Mesure NLL for in- and out-of-domain data. It seems to be still an adequate (indirect) criterion of calibration, and is an adequate criterion of uncertainty estimation. According to [Ashukha2020], the NLL needs a pre-calibrated model with temperature scaling for in-domain data (called calibrated NLL / calibrated LL).

b. To use the squared kernel calibration error (SKCE) proposed in [Widmann2019] along with de facto standard, but biased ECE. The SKCE is an unbiased estimate of calibration. There might be some pitfalls of this metric that I'm not aware of, but the paper looks solid and convincing. Also, please put attention to Figure 83 in the arХiv version.

Yes, ECE is the standard in the field, but it is the wrong standard that prevents us from meaningful scientific progress, so we should stop using it.

2) The standard deviation needs to be reported everywhere. Especially the differences between close values like (97.52, 97.52, 97.47 in Table 2) may appear not statistically significant. The same touches Fig 5 and other figures that are reported. Otherwise, it is impossible to stay how solid results are.

**Minor comments:**

1) Maybe it worth to provide plot mean-λi vs epoch to illustrate this "Notice that λi is dynamically updated at the end of each epoch."?

2) Figure 4(a) is done slightly disorderly.

3) In the paper, ECE is measured in percentages. As far as I can tell ECE is dimensionless quantities. It is not clear what is intended.

**Final comment:** I put the "marginally below acceptance threshold" score, but I'm willing to increase it after the update with corrections (and hope that these corrections will be done).  I like the direction and CAMixup, but in-domain results are not very consistent (see Fig 5 (d)), the ECE has uncontrollable model-specific biases that ruin all the presented results.

[Widmann2019] Widmann D, Lindsten F, Zachariah D. Calibration tests in multi-class classification: A unifying framework. In Advances in Neural Information Processing Systems 2019 (pp. 12257-12267). https://arxiv.org/pdf/1910.11385.pdf

[Ashukha2020] Ashukha A, Lyzhov A, Molchanov D, Vetrov D. Pitfalls of in-domain uncertainty estimation and ensembling in deep learning. ICLR, 2020.

---

> ### Author Response · Authors · 2020-11-18
> **We added new calibration metrics.**
>
> We would like to thank the reviewer for the feedback and suggestions to improve this paper. We address the reviewer’s concerns:
>
> ========================
>
> ECE is biased.
>
> ========================
>
> We would like to thank the reviewer for suggesting a more reliable metric. We ran additional experiments with the referred calibration metric SKCE [1] and the debiased calibration error in [2]. For SKCE, it takes 7 minutes to calculate the calibration error on CIFAR-10 testset (10,000 test samples). Therefore, we haven’t finished the CIFAR-100-C experiments (corruptions have 75 types and each has 3 random seeds). We revised the paper to include the results we have for now with SKCE and the debiased calibration error in Appendix F (Table 6 and Figure 12). We summarize some key results of SKCE on BatchEnsemble in the following:
>
> |  	|  	| BE 	| BE+Mixup 	| BE+CAMIxup 	| BE+AugMix 	| BE+AugMixup 	| BE+AugCAMixup 	|
> |:-:	|-	|:-:	|:-:	|:-:	|:-:	|:-:	|:-:	|
> | CIFAR-10 	  |  	| 3.48e-4 	| 4e-3 	| 1.3e-4 	| 8.48e-5 	| 2.4e-4 	| 1.54e-5 	|
> | CIFAR-100 	  |  	| 6.21e-4 	| 8.3e-3 	| 2.8e-4 	| 1.08e-3 	| 1.65e-3 	| 5.75e-4 	|
>
> As the results showed, Mixup still hurts the ensemble calibration under SKCE metric. The improvement of CAMixup over Mixup is even more obvious under the SKCE metric (roughly 30X reduction on both CIFAR-10 and CIFAR-100). The other calibration metric (debiased calibration error, figure 12 in the appendix F in the revision) also supports our conclusion.
>
> ========================
>
> Figure 5d is inconsistent.
>
> ========================
>
> The inconsistency happens between deep ensembles and deep ensembles with CAMixup. This is because Mixup hurts deep ensembles calibration much more than its degradation on BatchEnsemble or MC-dropout. Therefore, our proposed fix still has worse calibration than vanilla deep ensembles (only on testset, CAMixup is still much better than vanilla on corruptions). However, this inconsistency doesn’t change the main findings in the paper. The ranking of CAMixup and Mixup are consistent across all ensembles.
>
> ========================
>
> Other issues
>
> ========================
>
> We evaluated NLL averaged over 5 runs: BE+CAMixup outperforms BE+Mixup with 0.123 NLL vs 0.180 nats respectively. This is a significant improvement: a vanilla WRN-28-10 model obtains 0.159 nats / 96.0% accuracy and a vanilla BE without Mixup obtains 0.136 nats / 96.3% accuracy. This means BE+Mixup not only degrades calibration error but also degrades NLL. BE+CAMixup fixes this issue, achieving the best of all worlds over vanilla BE: accuracy, NLL, and ECE.
>
> Note we chose to evaluate NLL over temperature-scaled NLL. Ashukha et al. (2020) state: “LL demonstrates a high correlation with accuracy (ρ > 0.86), that in case of calibrated LL becomes even stronger (ρ > 0.95).” This means to evaluate uncertainty quality instead of predictive performance, LL may be a better measure. LL is also more well-known in theoretical studies of calibration forecasting, goodness-of-fit tests, and asymptotics.
>
> Standard error: We didn’t include the standard deviation in the main table for the presentation purpose. The closed numbers reviewer mentioned (97.52, 97.47) are the accuracy measure, which are the evidence that our proposed CAMixup doesn’t sacrifice accuracy much. Accuracy is not the main focus of this paper. We believe the difference in calibration error is significant enough. We will include the standard error in the next revision.
>
> ECE is measured in percentages for the space purpose. We also follow the style in [3]. For the newly added SKCE in Table 12, we dropped the percentages.
>
> [1]: Widmann, D., Lindsten, F., & Zachariah, D. (2019). Calibration tests in multi-class classification: A unifying framework. In Advances in Neural Information Processing Systems 2019.
>
> [2]: Kumar, A., Liang, P., & Ma, T. (2019). Verified Uncertainty Calibration.  In Advances in Neural Information Processing Systems 2019.
>
> [3]: Guo, C., Pleiss, G., Sun, Y., & Weinberger, K.Q. (2017). On Calibration of Modern Neural Networks. In International Conference on Machine Learning 2017.

---

> > ### Comment · AnonReviewer1 · 2020-11-24
> > **Thanks for the corrections, disagree on temperature-scaled NLL**
> >
> > > ECE is biased
> >
> > Thank you, that looks convincing. It is cool that an unbiased metric worked well.
> >
> > > Figure 5d is inconsistent
> >
> > That looks convincing; thank you for clarification.
> >
> > > Note we chose to evaluate NLL over temperature-scaled NLL.
> >
> > I appreciate the expressed opinion. The correlation of NLL/temperature-scaled NLL with Acc is indeed high. But it is important that high correlation does not mean that temperature-scaled NLL is a worse measure of uncertainty than NLL (I think it is still an open question which metrics are good and reflects all of the requirements to a model).
> >
> > In practice, if a model can be calibrated, there is no reason don't do it. We should clearly understand if the improvement we have is caused by a slightly better temperature of a model or the model is improved in a non-trivial way. Thus it is useful to compare models at the optimal temperature.
> >
> > > Standard error
> >
> > Worth adding to the appendix, at least.

---

> > > ### Author Response · Authors · 2020-11-25
> > > **Standard error is added in the latest revision**
> > >
> > > We would like to thank the reviewer for the reply! In the latest revision, we splited the old table 4 into the new table 4 and table 5 to include standard errors in the appendix.

---

### Official Review · AnonReviewer3 · 2020-10-28
**limited novelty and performance**

**Rating:** 4
**Confidence:** 4

**Review:**

Summary:

- This paper found that combining ensembles and data augmentation can harm model calibration. Inspired by this finding, it proposes a simple correction by only applying mixup to certain classes. Empirical experiments show some improvements.

Pros:
- The paper is clearly written and easy to follow. The motivation and approach is intuitive.
- Experiments are conducted on both small and large datasets.

Cons:
- The findings do not seem too surprising to me. The vanilla deep neural networks are often in the over-confident regime, so either ensemble or mixup itself seems very effective in improving model calibration. But applying these two together might lead the model into the under-fitting regime.
- It occurs to me that the proposed AugCAMixup is not good enough. From Table 2 we can tell that it basically sacrifices final accuracy with improvement in ECE. This means that applying mixup to all the classes is more preferrable than applyting to a subset of the classes in terms of accuracy. Why don't the authors consider using rescaling to fix the under confident issue?
- The baselines compared in this submission seem too limited. Some rescaling based methods should be considered, for instance, augmixup + rescaling.
- How's the performance on metrics other than ECE? I think it is important to include other metrics as ECE can hide some problems.

------
post-rebuttal update

I appreciate the authors for the responses. While the other reviewers give high reviews about this manuscript, I would keep my original reveiw for the following reasons. 1) This manuscript is incremental in nature. I agree with R3 that "understanding which techniques can be combined and which cannot (and how to fix it) is important for developing the field and feels like a small step to the right direction", but a somewhat unsurprising result lacks technical novelty. In one of the responses the authors wrote "The spirit of this work is to point out that not all data augmentations can be combined well with ensembles." I think this potentially means that the conclusion the authors made on mixup could not even transfer to other data augmentations. This even further limits the contribution of this manuscript. So I guess the above argument from R3 is not convincing, or one could try to study the effect of batch norm and ensemble and wrote another good paper. 2) The empirical performance of the method is limited and thus whether the proposed method is useful is a question.

---

> ### Author Response · Authors · 2020-11-18
> **Our work studies when data augmentations and ensembles have complementary benefits on calibration.**
>
> We would like to thank the reviewer for the feedback and suggestions to improve this paper. We address the reviewer’s concerns:
>
> Q: The findings do not seem too surprising to me. The vanilla deep neural networks are often in the over-confident regime, so either ensemble or mixup itself seems very effective in improving model calibration. But applying these two together might lead the model into the under-fitting regime.
>
> This intuition is not actually true, so we hope you do in fact find the conclusions surprising! Namely, Section 2 identifies that the compounding underconfidence only holds when combining ensembles with DA techniques which soften labels. This explains why ensemble methods have performed well in prior literature using more standard DA (e.g., random crops/flips) and recent SOTA techniques like AugMix. We provide more empirical evidence with this phenomena by showing ensembles+LS can harm calibration, and we provide a principled argument surrounding the conflation of model + data uncertainty.
>
> Q: It occurs to me that the proposed AugCAMixup is not good enough. From Table 2 we can tell that it basically sacrifices final accuracy with improvement in ECE.
>
> We’d like to point out that our contribution is the finding and analysis around the compounding underconfidence of marginalization techniques and DA methods which soften labels. To the best of our understanding, this is a novel and high impact finding: both ensembles and DA are presently the most powerful techniques in uncertainty benchmarks, and we find that they don’t always combine well.
>
> For Table 2 specifically, AugCAMixup does achieve roughly the same accuracy as AugMixup (97.47 v.s 97.52), so the sacrifice on the final accuracy is insignificant. This is a particularly notable achievement for new SOTA as AugCAMixup makes significant improvements on ECE: cutting error by half. And we successfully reduced the ECE on CIFAR-100c under 5% without hurting calibration on testset, which is a very strong result.
>
> Q: The baselines compared in this submission seem too limited.
>
> The spirit of this work is to point out that not all data augmentations can be combined well with ensembles. We made a detailed study on the underlying cause of the pathology. The main purpose of the experiment section is to show our proposed method is effective in fixing the pathology when combining Mixup and ensembles. This illustrates that we successfully identified the root cause of the pathology, suggesting that we should apply soft labels only on difficult examples instead of the entire training set, when combined with ensembles. Posthoc rescaling methods are orthogonal to our approach and do not fix the core issue with Mixup’s conflation of model+data uncertainty (Section 2). In Section 5, we show that the CAMixup is indeed complementary, combining well with TS for even better calibration. As [1] pointed out, rescaling doesn’t generalize well to distribution shift, which is an important evaluation metric in this work.
>
> Q: I think it is important to include other metrics as ECE can hide some problems.
>
> We agree that metrics other than ECE should be included. As R1 recommended, we include three additional metrics: SKCE/Debiased Calibration Error/NLL. Please see the reply to R1.
>
> [1]: Ovadia, Y., Fertig, E., Ren, J., Nado, Z., Sculley, D., Nowozin, S., Dillon, J.V., Lakshminarayanan, B., & Snoek, J. (2019). Can You Trust Your Model's Uncertainty? Evaluating Predictive Uncertainty Under Dataset Shift. In Advances in Neural Information Processing Systems 2019.

---

> ### Comment · AnonReviewer1 · 2020-11-24
> **Response to AnonReviewer3**
>
> > The findings do not seem too surprising to me.
>
> I would say that the effect should not necessarily be surprising, and I agree it is kind of expected here. But, understanding which techniques can be combined and which cannot (and how to fix it) is important for developing the field and feels like a small step to the right direction.
>
> > Why don't the authors consider using rescaling to fix the under confident issue?
> > The baselines compared in this submission seem too limited. Some rescaling based methods should be considered, for instance, augmixup + rescaling.
>
> I agree that temperature scaling is ignored in the paper. Even though, there is a temperature scaled NLL that sort of normalizing NLL on a rescaling performance. I mentioned it in my review too.

---

> ### Author Response · Authors · 2020-11-24
> **Our work studies when data augmentations and ensembles have complementary benefits on calibration.**
>
> Thank you again R3 for providing detailed feedback! If you still have specific concerns, please let us know. This is the last day for us to reply, and we hope we can best address any lingering questions.

---

### Public Comment · ~Jize_Zhang1 · 2020-11-16
**Related work on ensemble + calibration**

Great work on calibrating ensembles! I also want to refer to our recent work on improving post-hoc calibration methods using ensembles [1]. In addition, we also provided a binning-free KDE-based estimator to reduce the bias and binning sensitivity issues of existing histogram ECE estimators. All codes are also available online.

[1] Jize Zhang, Bhavya Kailkhura, and T Han. "Mix-n-Match: Ensemble and compositional methods for uncertainty calibration in deep learning.", ICML 2020, https://arxiv.org/pdf/2003.07329.pdf

---

> ### Author Response · Authors · 2020-11-19
> **Thanks for suggesting a new calibration metric!**
>
> Hi Jize,
>
> We would like to thank you for suggesting another calibration metric which has less bias than ECE and open-sourcing the code. We used your ece_kde_binary function on our CIFAR-10 experiments. It gives the same ranking as SKCE and debiased calibration error which we added in the revision. We will include the result with KDE-based calibration estimator in the next revision.

---

### Decision · Program_Chairs · 2021-01-07
**Final Decision**

**Decision:**

Accept (Poster)

**Comment:**

This paper analyses the interaction between data-augmentation strategies  and model ensembles with regards to calibration performance. The authors note how strategies such as mixup and label smoothing, which reduce a single model's over-confidence, lead to degradation in calibration performance when such models are combined as an ensemble. They propose a simple solution. The paper merits publication.